evolution

determinate growth, endothermy, indeterminate growth, growth plate cartilage, micro-CT, squamata

**Authors for correspondence:**
Petra Frýdlová
e-mail: petra.frydlova@seznam.cz
Daniel Frynta
e-mail: frynta@centrum.cz

†Joint senior authors.

# Determinate growth is predominant and likely ancestral in squamate reptiles

Petra Frýdlová[1,3], Jana Mrzílková[2,3], Martin Šeremeta[2,3], Jan Křemen[2,3], Jan Dudák[4], Jan Žemlička[4], Bernd Minnich[5], Kristina Kverková[1], Pavel Němec[1,†], Petr Zach[2,3,†] and Daniel Frynta[1,†]

[1]Department of Zoology, Faculty of Science, Charles University, Prague 12844, Czech Republic
[2]Specialized Laboratory of Experimental Imaging, Third Faculty of Medicine of Charles University, Institute of Technical and Applied Physics and Faculty of Bioengineering, Prague 100 00, Czech Republic
[3]Department of Anatomy, Third Faculty of Medicine, Charles University, Prague 100 00, Czech Republic
[4]Institute of Experimental and Applied Physics, Czech Technical University in Prague, Prague 110 00, Czech Republic
[5]Department of Biosciences, University of Salzburg, Hellbrunnerstrasse 34, Salzburg 5020, Austria

PF, 0000-0001-9385-9743; JM, 0000-0002-9474-5040; JD, 0000-0002-4064-1246; JŽ, 0000-0002-4816-4827; KK, 0000-0002-4579-6861; PN, 0000-0003-0277-0239; PZ, 0000-0003-1941-2972; DF, 0000-0002-1375-7972

Body growth is typically thought to be indeterminate in ectothermic vertebrates. Indeed, until recently, this growth pattern was considered to be ubiquitous in ectotherms. Our recent observations of a complete growth plate cartilage (GPC) resorption, a reliable indicator of arrested skeletal growth, in many species of lizards clearly reject the ubiquity of indeterminate growth in reptiles and raise the question about the ancestral state of the growth pattern. Using X-ray micro-computed tomography (μCT), here we examined GPCs of long bones in three basally branching clades of squamate reptiles, namely in Gekkota, Scincoidea and Lacertoidea. A complete loss of GPC, indicating skeletal growth arrest, was the predominant finding. Using a dataset of 164 species representing all major clades of lizards and the tuataras, we traced the evolution of determinate growth on the phylogenetic tree of Lepidosauria. The reconstruction of character states suggests that determinate growth is ancestral for the squamate reptiles (Squamata) and remains common in the majority of lizard lineages, while extended (potentially indeterminate) adult growth evolved several times within squamates. Although traditionally associated with endotherms, determinate growth is coupled with ectothermy in this lineage. These findings combined with existing literature suggest that determinate growth predominates in both extant and extinct amniotes.

## 1. Introduction

Growth patterns can be classified as either determinate or indeterminate [1]. Determinate growth ceases during the natural lifespan of individuals. Growth trajectory and adult body size are primarily genetically determined but may, to some degree, be influenced by the environment [2]. Indeterminate growth, by contrast, continues throughout the life of an individual. Adult body size is not genetically determined and growth trajectory, as well as growth rate, retains the lifelong ability to change with environmental conditions [2]. However, these general definitions encompassing a wide range of growth types of both vertebrate and invertebrate animals do categorize extended adult growth of amniotes, which is commonly referred to as indeterminate growth in the vertebrate literature, as attenuating determinate rather than indeterminate growth [2]. Therefore, here we adopt a less stringent definition introduced by Karkach postulating that 'growth is determinate if an organism reaches maximum (asymptotic) size when many individuals of the population are still

alive, and indeterminate when very few individuals are alive' [3]. This approach stresses quantitative rather than qualitative differences between growth patterns.

Although it is often expected that determinate growth stops at, or soon after, reaching sexual maturity, it may continue long past this age [3]. To complicate matters further, indeterminate growers may exhibit asymptotic growth, provided that the environment imposes energetic constraints on their growth, and determinate growers do not have to achieve asymptotic growth due to high mortality in natural populations. Thus, the practical distinction between these two types of growth may be difficult on the basis of empirical growth curves alone, especially in cases when indeterminate growth greatly decelerates with age and/or determinate growth persists after sexual maturity and its rate is modulated by the environment [2,3]. Therefore, it is desirable to validate long-term growth studies with osteological examinations, which not only have the potential to reveal irreversible arrest of skeletal growth and thereby provide conclusive evidence for determinate growth but also can do so relatively quickly across large numbers of species representing various clades.

Endothermic tetrapods are traditionally described as determinate growers [4–6], whereas ectothermic vertebrates as indeterminate growers [4,7–10]. This paradigm is commonly mentioned in textbooks and scientific papers [11–20], although exceptions abound (e.g. indeterminate body growth in kangaroos [21], determinate body growth in fishes [22,23] and reptiles, see below). The notion of indeterminate growth being prevalent in reptiles remains popular despite accumulating evidence for determinate growth [24–31]. The idea that reptiles grow throughout life seems to be perpetuated by long lifespans of many species [32], slow growth that continues after sexual maturity [33], and growth rates largely dependent on environmental conditions [34,35].

Evidence for determinate growth in reptiles comes from two independent lines of research. First, numerous capture–recapture field studies (e.g. [25,36–39]), as well as laboratory experiments (e.g. [40,41]), involving distantly related species of reptiles suggest that growth is minimal or non-existent in older individuals. Second, osteological studies show that distantly related reptiles attain skeletal maturity associated with the arrest of skeletal growth. Before we review the osteological evidence, let us briefly summarize the mechanisms underlying the growth of long bones in amniotes.

Endochondral and periosteal bone formation underpin bone growth in length and thickness, respectively [12]. Due to mechanical reasons and shared molecular regulation, these two types of bone formation are coupled and synchronized [12,42]. Even mature bones, however, have a capacity for remodelling, which is often associated with periosteal apposition of new lamellar bone, even after the cessation of longitudinal growth [43]. Longitudinal bone growth depends on growth plate cartilages (GPCs). These are proliferative zones located in long bones, ribs and vertebrae [7], responsible for cartilage formation and its subsequent endochondral ossification [44–46]. When chondrocyte proliferation ceases and the GPC disappears, longitudinal skeletal growth irreversibly arrests [12,47]. Bone growth in girth is mediated by synergic activity of the endosteum and the periosteum. While osteoclasts present in the endosteum resorb the bone from the inside to prevent it from becoming unnecessarily thick, osteoblasts present in the periosteum secrete bone matrix and induce its intramembranous ossification, resulting

in apposition of new layers of lamellar osseous tissue on the surface of the bone diaphysis [12,43]. Seasonal periods of slow periosteal growth are discernible on a transverse bone section as rings of highly mineralized compact bone, called lines of arrested growth (LAGs). Substantial attenuation of periosteal growth typically leads to the development of the external fundamental system (EFS), which is a microstructure of closely spaced series of LAGs at the outermost margin of the bone diaphysis [12]. The presence of the EFS thus indicates cessation of any significant circumferential growth of a bone. Synchronous timing of GPC degradation, EFS development and body growth arrest has been recently demonstrated [48]. Taken together, both the absence of the GPCs and presence of the EFS are unambiguous signs of skeletal maturity and therefore reliable markers of determinate growth.

Radiographic and histological examinations have demonstrated the absence of GPCs in long bones of small species of monitor lizards [26], a finding that we have independently corroborated using micro-computed tomography (µCT) and micro-radiography (µRTG) [30]. The EFS has been reported (or can be recognized in published microphotographs) in the tuatara [24], the Mosor rock lizard [49], eight species of iguanas [28], the American alligator [27,50] and the leopard tortoise [51]. The EFS has also been reported in representatives of fossil archosaurs, including crocodilians [27,52], pterosaurs [53,54], non-avian dinosaurs [55,56], and in subfossil and recent birds [57,58]. These data provide clear evidence that at least some reptiles are determinate growers. More extensive osteological examinations across a broad array of taxa are needed to determine how widespread determinate growth is within both extinct and extant reptiles.

To address this issue in squamate reptiles, we have recently examined the presence/absence of GPCs in the femur of adult individuals of 85 species from several lizard lineages, namely monitor lizards and their relatives [30] and iguanians [31], using µRTG and µCT. These investigations revealed the absence of GPCs, implying determinate growth, in most of the examined adult iguanas and small-bodied monitor lizards. By contrast, GPCs were present in most adult agamas, chameleons and large-bodied monitor lizards, suggesting they exhibit extended (potentially indeterminate) growth.

Here, we focus on more basally diverged squamate lineages: geckos (Gekkota); skinks, plated and girdle-tail lizards (Scincoidea); wall lizards, whiptails and tegus (Lacertoidea). We categorized 85 species from these clades as either determinate or potentially indeterminate growers, based on femoral GPC states determined from µCT scans and/or µRTG. We apply a working definition of determinate growth, which is very restrictive. We consider determinate growers only those species, in which adult individuals exhibit a clear sign of irreversible arrest of longitudinal skeletal growth, i.e. fully resorbed GPCs. Thus, when animals are still capable of growth, although the actual growth rates might be negligible, we consider them potentially indeterminate growers. While this approach may overestimate the number of species exhibiting indeterminate growth, identification of determinate growers is conclusive. Moreover, to map out the evolution of growth type within Lepidosauria, we performed ancestral state reconstruction using a dataset of 164 species representing all major clades of lizards and the tuataras. The results of our study indicate that determinate growth is predominant and likely ancestral and that extended

(potentially indeterminate) adult growth evolved several times within squamate reptiles.

## 2. Material and methods

Altogether, we examined femoral bones of 194 individuals of 85 lizard species representing three basal clades of Squamata, namely Gekkota, Scincoidea and Lacertoidea (electronic supplementary material, table S2). We examined mainly adult individuals, but several subadults were included for reference as well. To obtain adult individuals of known age, the majority of samples were taken from captive bred lizards. Specifically, out of 194 individuals examined, 130 were captive bred and 64 wild animals (the origin of samples is labelled in electronic supplementary material, table S2 as captive versus wild). Samples originated mostly from collections of the Charles University Department of Zoology and Prague Zoo. For the reconstruction of ancestral states, we added data from previous studies [26,30,31].

Since body mass is largely dependent on body condition and significant mass gains can occur even in adults with arrested skeletal growth, body length was taken as a proxy for body size. Prior to analysis, we measured the snout–vent length to the nearest 0.1 mm and expressed it as an absolute (SVL) and relative ($SVL_{rel}$) value. The latter represents a per cent ratio of SVL of the examined specimen to the maximum SVL reported in the literature for the particular species and sex; i.e. $SVL_{rel}$ 100% means the maximum reported size. The data for absolute and relative SVLs are summarized in electronic supplementary material, table S2, $SVL_{max}$ and references in dataset 1).

We analysed GPC in the proximal part of the femur by μRTG and μCT. The bone was dissected, mechanically cleaned and examined using either a Bruker SkyScan 1275 μCT scanner or a custom-built μCT system designed for small animal imaging [59,60], following previously described procedures [30,31]. Epiphyseal senescence and ossification status were evaluated blindly by two independent observers. The presence of the GPC was clearly visible on μRTG and μCT scans as fine-grained cartilage and a suture between the metaphysis and epiphysis; a relatively dense structure, likely associated with the secondary ossification centre, was typically recognizable inside the epiphysis. By contrast, trabecular bone typically fills up the entire metaphysis and epiphysis in bones with a fully resorbed GPC. The inner structure of the bone was assessed in detail using 3D visualizations made from μCT scans to rule out the presence of the cartilage and/or suture remnants between the metaphysis and the epiphysis. We scored the GPC state in a binary fashion, as either present (1) or absent (0). In several cases, we detected the process of GPC degradation. The GPC was less clearly visible, nearly resorbed, but still present to some extent. This stage was also coded as GPC present.

Ancestral state reconstruction was performed in R using the Hidden State Speciation and Extinction (HiSSE) model implemented in the R package hisse [61], to allow for different transition rates between states and different diversification rates. Additionally, we performed the ancestral states reconstruction using maximum parsimony, as implemented in the R package castor [62]. The reconstructed states were plotted using the R package phytools [63]. Only species where we had adult individuals with $SVL_{rel}$ greater than 80% were included in the ancestral state reconstruction analysis (see Dataset 2) since previous research on body growth revealed that squamate reptiles typically attain sexual maturity close to this relative body size [33]. However, this dataset might still be biased towards indeterminate growth, since we cannot exclude false-positive results (i.e. the presence of GPC in animals, which were not fully grown and will stop their growth later in ontogeny).

Further methodological details and discussion of potential technical limitations are provided in the electronic supplementary material.

## 3. Results

Using high-resolution μRTG and μCT, we determined the presence/absence of epiphyseal growth plates in the femoral epiphysis. We analysed femoral bones from 85/194 species/individuals of lizards (Gekkota (38/100), Scincoidea (26/53) and Lacertoidea (21/41)). We found both states of GPC, fully developed (figure 1a,c,e; electronic supplementary material, video file S1, 2), as well as fully resorbed (figure 1b, d,f; electronic supplementary material, video file S3, 4), and a few cases of ongoing GPC degradation. GPC was present in juveniles, subadults and some adults (electronic supplementary material, table S2). Nevertheless, in most adults, the GPC was fully resorbed. However, some plasticity exists, and different individuals of the same species, sex and size can differ in GPC state. We took the absence of the GPC in any individual as evidence of determinate growth in a given species, even if the GPC was preserved in other adults of that species since it shows there is a point where growth will eventually stop. To outline the growth pattern across all Lepidosauria, we used data gathered in this study, datasets from our previous studies [30,31] and earlier data showing evidence of determinate growth in the monitor lizards [26] and tuataras [24,64,65]. We plotted the GPC state (present/absent) on a phylogenetic tree (figure 2). It is evident that GPC degradation, implying irreversible arrest of growth, is common in all major clades. Reconstruction of the ancestral states using both parsimony (electronic supplementary material, figure S1) and likelihood criteria suggests determinate growth type in the ancestors of Gekkota, Scincoidea and Lacertoidea (figure 2; electronic supplementary material, figures S2, S3). Preserved GPCs, suggesting extended, potentially indeterminate adult growth, appear independently several times, e.g. in chameleons and agamas, large species of monitor lizards, and teiids. Furthermore, both parsimony and likelihood models agree that the squamate ancestor was likely a determinate grower (with 85–88% probability in likelihood models; see electronic supplementary material for more details).

Although it was not the focus of this study, we also performed histological examinations on a limited sample to assess whether the arrest of longitudinal bone growth is associated with arrest of bone growth in girth and indeed observed tightly spaced rings of laminar bone depositions in the outer bone cortex, a clear indication of decelerated or ceased periosteal growth, in individuals with fully resorbed GPCs (electronic supplementary material, figure S4; see results in electronic supplementary material for details).

## 4. Discussion

The μRTG and μCT examinations performed in this study revealed that the majority (59 out of 75) of the examined species representing three basal clades of squamate reptiles, namely Gekkota, Scincoidea and Lacertoidea, exhibit entirely resorbed femoral GPCs and are therefore likely determinate growers. Determinate growth is not only predominant in all these clades but also probably ancestral. Moreover, the data gathered here and in previous studies [26,30,31] strongly

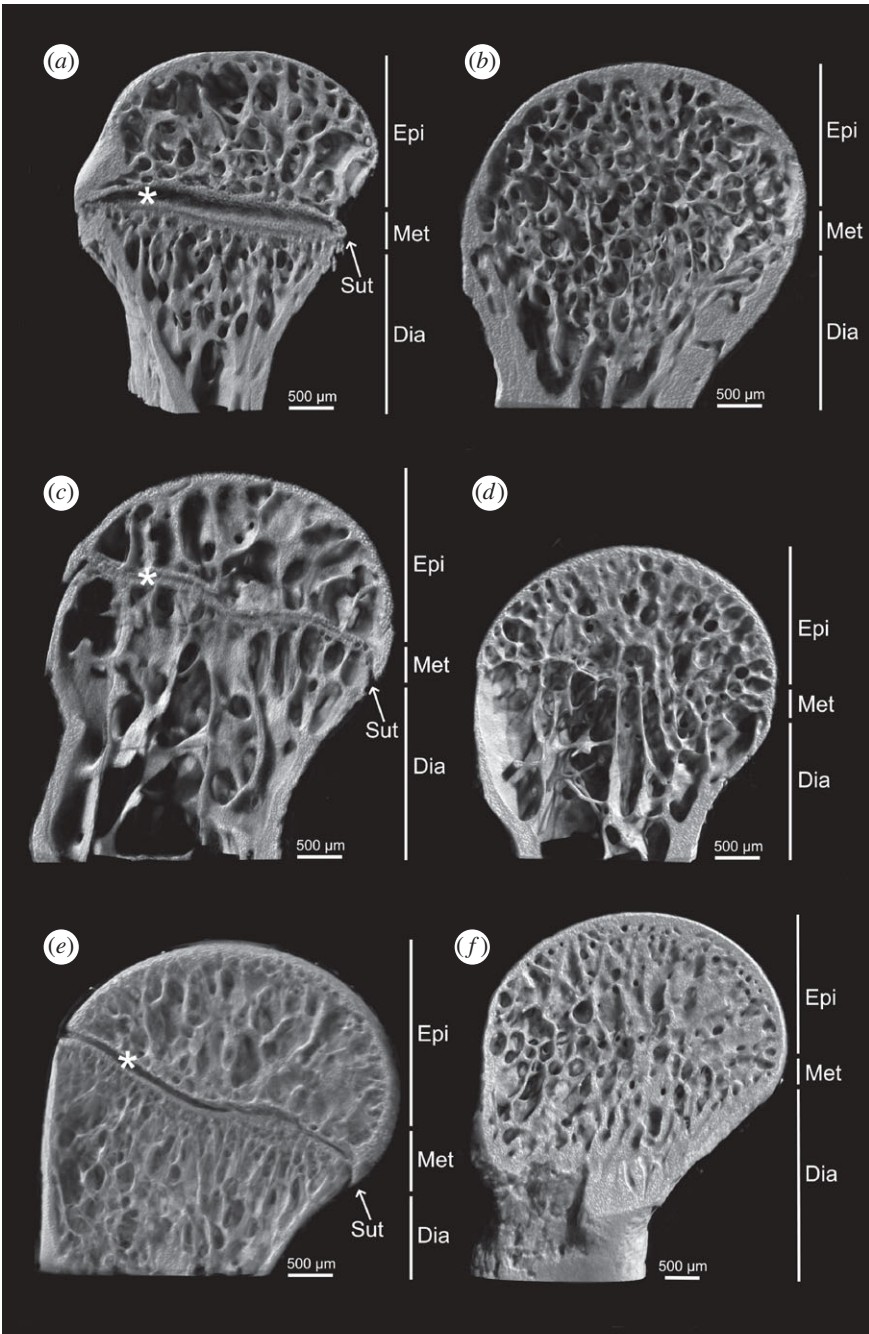

**Figure 1.** Epiphyseal growth plate cartilage visualization. Longitudinal section of the proximal part of the femur by μCT in three representative species of basal clades (Gekkota, Scincoidea and Lacertoidea). The epiphyseal growth plate is present in a subadult (*a*) and absent in an adult of the crested gecko *Correlophus ciliatus* (*b*); present in a subadult (*c*) and absent in an adult of Karsten's girdled lizard *Zonosaurus karsteni* (*d*), present in a subadult (*e*) and absent in an adult of Moroccan eyed lizard *Timon tangitanus* (*f*). Asterisks indicate the growth plate cartilages. Abbreviations: Dia, diaphysis; Epi, epiphysis; Met, metaphysis; Sut, suture. Note the completely different inner structure in the metaphysis in *a, c, e* and *b, d, f*. Scale bars, 500 μm.

suggests that the same applies for all squamate reptiles. In total, we observed resorbed femoral GPCs in 106 out of 164 lizard species. The reconstruction of character states shows that determinate growth is likely ancestral for the entire clade of Squamata and that extended (potentially indeterminate) adult body growth probably evolved several times within squamate reptiles, most notably in chameleons and agamas, large species of monitor lizards, and teiids. Available literature data suggest that tuataras, crocodiles, birds and at least some turtles are determinate growers as well (for details see below). This is in stark contrast with the long-held view that indeterminate growth is the ancestral condition and is predominant in reptiles (reviewed in [16]). As argued in

detail below, determinate body growth has evolved early and predominates in both sauropsid and synapsid amniotes.

## (a) Determinate versus indeterminate body growth

An important, yet unanswered question is whether determinate and indeterminate body growth in amniotes involve two different regulatory mechanisms or are simply the result of a difference in timing (heterochrony) of GPC degradation, as we have suggested earlier [30]. Although some differential sensitivity of the GPC to steroid hormones is to be expected, the frequency of evolutionary transitions between these two growth types observed in squamate reptiles

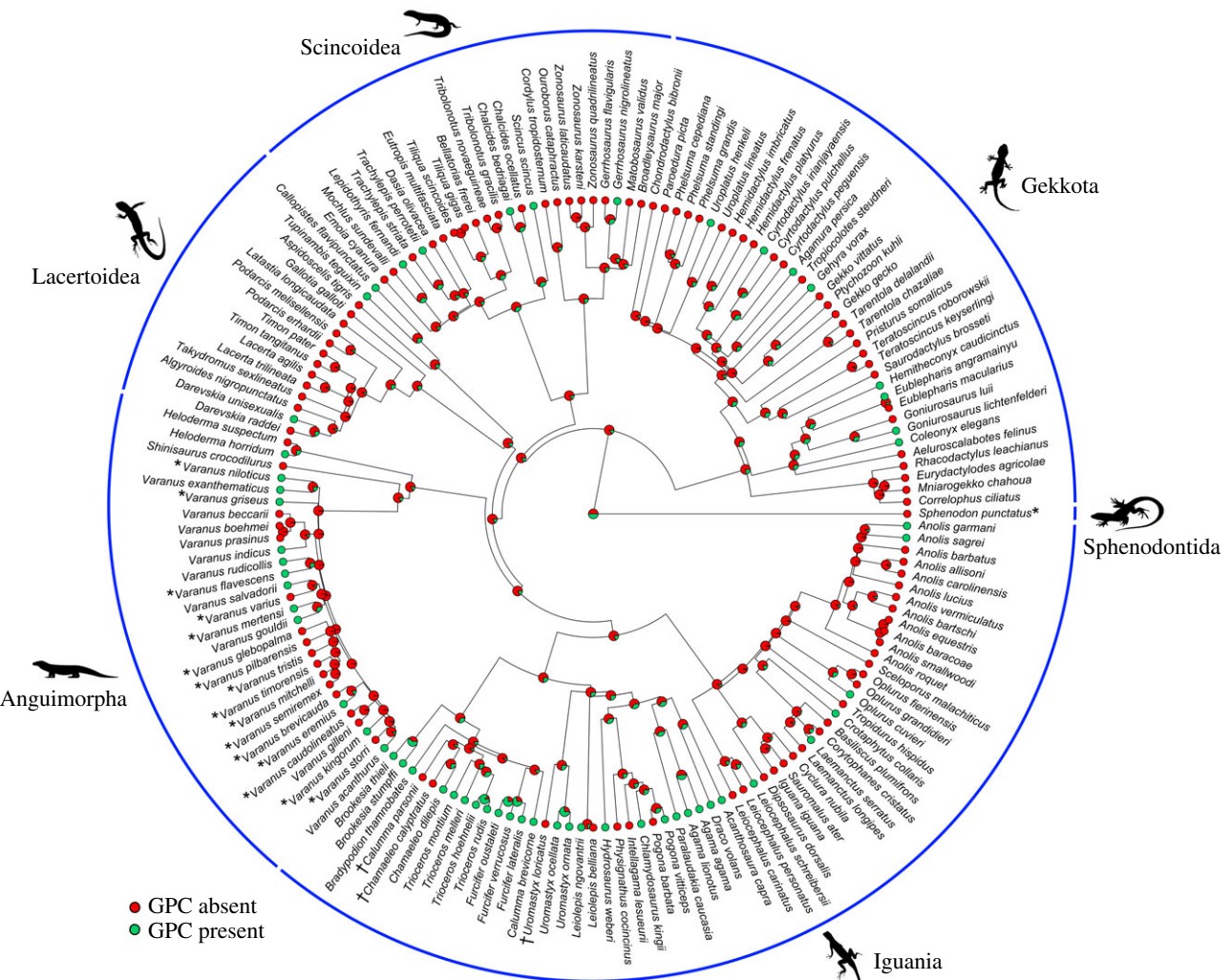

**Figure 2.** Ancestral state reconstruction of growth type in squamates. A circular tree depicting the growth plate cartilage (GPC) state in whole Squamata as revealed by μCT examination of the proximal part of femoral bones. Ancestral state reconstruction using maximum-likelihood with hidden state speciation and extinction models was employed to uncover the evolution of growth type (determinate versus indeterminate) in Squamata. The plotted ancestral states are reconstructed under the best model based on AIC (HiSSE equal rate, see electronic supplementary material, for more details). Presence and absence of the GPC imply extended (potentially indeterminate) and determinate body growth, respectively. Tuatara (*Sphenodon punctatus*), as a sister group of Squamata, was included as an outgroup. The state of tuatara is according to the presence of an external fundamental system and recapture growth data suggesting the determinate type of body growth [24,64]. Species marked with asterisk were scored according to the GPC state from the literature [26]. Species marked with † were very old individuals (for details of age see [30,31]).

(cf. Figure 2) clearly speaks in favour of a change affecting the timing of endochondral ossification. The very fact that the GPC disappears in extremely senescent individuals (e.g. 24-year-old *Varanus indicus* [30]; 30-year-old *Uromastyx loricatus* [31]) belonging to reptile groups that otherwise feature extended (potentially indeterminate) adult growth is also in line with the hypothesis that it is the timing that sets apart determinate from indeterminate growth, at least in squamate reptiles. If true, these growth patterns constitute different life-history strategies rather than distinctly regulated growth mechanisms. At one extreme of the determinate–indeterminate growth continuum are determinate growers that preferentially allocate energy to growth in order to reach a final size as soon as possible (e.g. [66]), at the other extreme are species that delay skeletal maturity to such an extent that longitudinal growth cessation and GPC resorption is seldom, if ever, observed in natural populations (e.g. [67]). Such a quantitative distinction between growth types is in accordance with the definition introduced by Karkach [3].

### (b) Determinate growth in reptiles and other amniotes

Long-term studies are the only way to accumulate quantitative data on individual growth. However, it is challenging to maintain these studies over sufficiently long time periods to encompass the natural lifespan of individuals, especially in large reptiles that tend to be long-lived. Consequently, long-term growth data are available only for a limited number of species (e.g. [25,36–39,41,68,69]) and do not allow the reconstruction of growth type evolution across reptiles. In the absence of these data, osteological evidence is invaluable.

Epiphyseal senescence and ossification status in reptiles have received only limited attention, although detailed studies in mammalian models are available (e.g. [70–72]). Complete resorption of the GPC has been reported for the first time in small species of monitor lizards [26]. We have corroborated these findings [30] and subsequently demonstrated resorption of the GPC in 106 species distributed across squamate reptiles, while also consistently finding preserved GPCs in chameleons and agamas ([31], present study).

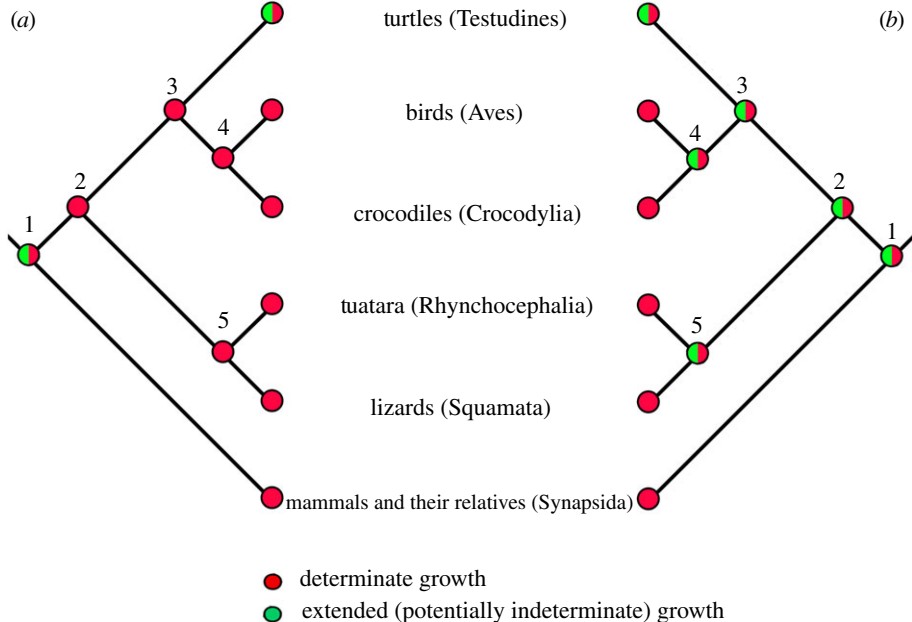

**Figure 3.** Two scenarios for the evolution of body growth type in amniotes. (*a*) Scenario based on maximum-parsimony predicting that the last common ancestors of the Lepidosauria (node 5), Archosauria (4), Archelosauria (3) and Sauropsida (2) were determinate growers. (*b*) More conservative scenario considering the possibility of multiple evolutionary transitions between determinate and extended (potentially indeterminate) growth on long branches does not allow conclusive inferences about the ancestral growth type for the above-mentioned nodes. Importantly, this ambiguity should not be taken as evidence in favour of extended (potentially indeterminate) growth. Both scenarios are inconclusive as far as the last common ancestor of the Amniota (1) is concerned, due to lack of data on growth type in the most basal sauropsids and synapsids. Groups were scored according to the presence/absence of growth plate cartilage, external fundamental system and capture–recapture field studies. See text for details.

Whereas chameleons are typically short-lived, and it could be argued that the process of GPC degradation takes longer than their natural lifespan, agamas can be long-lived and still preserve femoral GPCs well into adulthood (for more details see [31]). In line with our observations, recent histological studies have reported complete resorption of the GPC in the gecko *Paroedura picta* [73] and the teiid lizard *Aspidoscelis tigris* [74], but no sign of GPC degradation in the agama *Stellagama stellio* [67]. Moreover, EFS was demonstrated in iguanid lizards [28]. Together, these results suggest that determinate growth is widespread among extant lizard species.

The ancestral state reconstruction performed here suggests that the determinate type of body growth is ancestral for squamate reptiles. Moreover, there is evidence of determinate growth in the extant species of tuatara, representing the sister group of squamate reptiles, Rhynchocephalia. EFS was demonstrated in adults [24] and some individuals in capture–recapture studies showed no signs of growth in 30 years [24,64]. Thus, the last common ancestor of Lepidosauria might also have been a determinate grower (figure 3). This hypothesis is supported by the parsimony model (electronic supplementary material, figure S1), but likelihood models remain inconclusive (50–59% probability, figure 2; electronic supplementary material, figures S2 and S3), reflecting the possibility of multiple evolutionary transitions between determinate and indeterminate growth on this long branch.

As mentioned above, determinate growth is well established in archosaurs (Archosauria), comprising extant crocodiles, birds and several fossil groups including dinosaurs (Dinosauria). Radiographic studies demonstrated complete resorption of the GPC and fusion of the primary and secondary ossification centres in long bones of birds [75–77]. The EFS has been reported in many archosaurs, including subfossil birds [58], non-avian dinosaurs (e.g. [55,56,78]), pterosaurs

(e.g. [53,54]), crocodilians [27,50] and their fossil relatives [52,79]. Moreover, determinate growth in the American alligator has been corroborated by long-term recapture data [37].

The situation is less clear in turtles (Testudines), the sister group to Archosauria. Long-term growth studies, complicated by the extreme longevity of these animals, produced contradictory results. Some species are reported to be indeterminate growers (*Chelydra serpentina* [37]; *Chrysemys picta* [80]), while others seem to have determinate body growth (*Emydoidea blandingi* [25]; *Chelonia mydas* and *Caretta caretta* [38]). A recent study brought evidence of EFS in *Stigmochelys pardalis*, supporting determinate body growth in turtles [51]. The current state of knowledge does not allow inferences about the ancestral growth type for turtles.

The EFS has been also demonstrated in Triassic reptiles belonging to the groups Placodontia [81], Allokotosauria and Phytosauria [82]. However, to our knowledge, almost nothing is known about more basal sauropsid reptiles. Two to three closely spaced LAGs have been observed in the outer cortex of a single humerus of the basal eureptile *Captorhinus* that lived during the Permian period [82].

Mammals, the only extant synapsids (Synapsida), constituting a sister group of the sauropsid reptiles, are determinate growers as evidenced by both long-term growth studies [83–85] and osteological data (e.g. [6,48,86]). Until recently, it was assumed that early synapsids exhibited indeterminate growth and a mammalian-like growth pattern evolved later in Therapsida (reviewed in [87]). However, osteological data gathered over the past decade clearly show that diverse pelycosaurs (Pelycosauria, a paraphyletic group of early non-therapsid synapsids) were also determinate growers—EFS has been reported in several genera representing at least three pelycosaur clades [88–91]. Yet, osteological features

implying determinate growth have not been observed in the most basal pelycosaurs [92,93]. The same holds true for Diadectes representing Diadectomorpha, a putative sister group of amniotes [92].

Taken together, the osteological evidence reviewed above strongly suggests that determinate growth predominates in both extant and extinct amniotes. However, current knowledge does not permit the ancestral growth pattern for amniotes to be determined unequivocally (figure 3). Further comparative scrutiny with osteological and/or modern imaging techniques, especially data on stem mammals, stem sauropsid reptiles and turtles, is needed to clarify this issue.

### (c) Decoupling between endothermy and determinate body growth

A high and stable body temperature, or homeothermic endothermy, has convergently evolved in birds and mammals (e.g. [94,95]). Endothermy is coupled with a high basal metabolic rate and fast growth that typically considerably slows down or ceases at sexual maturity [96]. Therefore, endotherms are defined as determinate growers, while ectotherms are traditionally considered indeterminate growers (see Introduction). Here, we show that such a clear-cut distinction is not tenable, because determinate growth is coupled with ectothermy in many lineages of squamate reptiles.

Timing of the evolutionary origin of endothermy is complicated by the fact that physiological processes do not fossilize. Nasal turbinates, associated with endothermy, have been found in members of the synapsid lineage from the Late Permian and Early Triassic periods (250–200 Ma), suggesting that synapsids had increased metabolic rates approximately 30–40 Myr before the emergence of true mammals [94]. Histology of long bones (fibrolamellar bone) suggests that endothermy and fast skeletal growth has evolved even earlier, specifically in ophiacodontid pelycosaurs (Ophiacodontidae) during the Early Permian or perhaps Late Carboniferous [91,92].

The situation is less clear in the sauropsid lineage. Early birds and theropod dinosaurs do not seem to have had nasal turbinates, however, based on structural (fibrolamellar bone, (proto)feathers), physiological (concentration of $O^{18}$ deposited in teeth) and behavioural traits (fossil evidence of egg incubation), the origin of endothermy in this lineage can be traced to the Jurassic period, before the diversification of theropod dinosaurs (for review, see [97]). Some authors suggest that endothermy evolved already in the Triassic period, at the beginning of the diversification of archosaurs

[98,99]. In any case, the data presented here suggest that the origin of determinate growth predated the origin of endothermy in the sauropsid lineage.

## 5. Conclusion

The absence of the femoral GPC, indicating determinate body growth, was the predominant pattern found in squamate reptiles. The reconstruction of character states suggests that determinate growth is ancestral for all squamate reptiles and that extended (potentially indeterminate) adult body growth evolved several times within squamates, likely as a result of heterochrony. If we include data from the literature demonstrating determinate growth in tuataras and archosaurs, it is clear that determinate body growth is common across sauropsid reptiles (figure 3). Together, these findings bring strong evidence against the assumption that all ectothermic vertebrates are indeterminate growers.

Data accessibility. Data used in this study are available from the Dryad Digital Repository: https://doi.org/10.5061/dryad.dbrv15dxz [100].

Authors' contributions. P.F. and D.F. conceptualized and designed the project. P.F., P.N., K.K. and D.F. provided the material. P.F. was responsible for data collection and curation. P.F., J.M., M.Š., J.K., B.M., J.D. and J.Ž. were responsible for performing micro-CT scans and data processing. P.F., D.F. and K.K. analysed the data. P.F. prepared the figures. J.M., M.Š., J.K., J.D., J.Ž., P.N. and P.Z. obtained funding acquisition. P.F., D.F., P.N. and K.K. wrote the original manuscript. All authors reviewed the draft.

Competing interests. We declare we have no competing interests.

Funding. This project was supported by the European Regional Development Fund Project 'Engineering applications of microworld physics' (no. CZ.02.1.01/0.0/0.0/16_019/0000766) and the Czech Science Foundation (project no. 18-15020S). The participation of P.F. was supported by the Charles University Research Centre program no. 204069.

Acknowledgements. We thank Nataša Velenská, Petr Velenský and Ivan Rehák (Prague Zoo), Jiří Moravec (National Museum), Jiří Marek (Zoopark Zájezd), Jan Konáš and Tomáš Jirásek (Zoological and Botanical Garden Pilsen), Antonín Hnízdil (Zoological Garden and Terrarium Prague), Jan Hříbal (Stanice přírodovědců DDM Prague), Petr Kodym (National Institute of Public Health), Daniel Koleška (Czech University of Life Sciences), Veronika Zahradníčková (Prague Zoo), Kristýna Sedláčková (Charles University), Kristýna Šifnerová (Institute of Geology of the Czech Academy of Sciences), František Šubík (Zoological Garden Ústí nad Labem) and Luděk Pokorný for providing specimens. We are grateful to Martin Kocourek (Charles University) for assistance with dissections, Milada Halášková and Renata Peterková (Charles University) for help with histological methods and Barbora Straková (Charles University) for helping us with circular tree visualization. We also thank two anonymous reviewers for helpful feedback on the manuscript.

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
