## [Reviewer comments · Proceedings of the Royal Society B: Biological Sciences]

Review History

RSPB-2020-0670.R0 (Original submission)

Review form: Reviewer 1

Recommendation

Major revision is needed (please make suggestions in comments)

Scientific importance: Is the manuscript an original and important contribution to its field?

Good

General interest: Is the paper of sufficient general interest?

Good

Quality of the paper: Is the overall quality of the paper suitable?

Good

Is the length of the paper justified?

Yes

Should the paper be seen by a specialist statistical reviewer?

No

Do you have any concerns about statistical analyses in this paper? If so, please specify them explicitly in your report.

No

It is a condition of publication that authors make their supporting data, code and materials available - either as supplementary material or hosted in an external repository. Please rate, if applicable, the supporting data on the following criteria.

Is it accessible?

Yes

Is it clear?

Yes

Is it adequate?

No

Do you have any ethical concerns with this paper?

No

Comments to the Author

The authors present a detailed analyses of the Growth Plate Cartilage (GPC) resorption using uCT on a wide range of taxa within the Lepidosauria as a means to see if there is in fact indeterminate growth within this group.

Overall I found their uCT interpretation sound, though I would have liked to see some more images that included some more detailed histology (higher resolution). However, I have two main concerns. The first is their definition of "indeterminate growth" for vertebrates, which MUST be addressed for this paper to make any sense. The second is to talk more about their use of captive animals. See below for more details.

1) What is the definition of indeterminate growth, specifically for vertebrates? From what I had understood, indeterminate growth just meant perpetual growth. Thus this should include both growth plate lengthening AND periosteal enlargements. In fact, most of what I've seen for indeterminate growth in vertebrates follow the later form of perpetual growth that is "width-wise" through the periosteum, which could continue AFTER the closure of the growth plates. In paleohistology, we usually look for the formation of an Outer Circumferal Layer (OCL) to mark the end of growth.

In addition, it would be very messy to use the loss of the growth plate cartilage as your definition of arrested growth as rodents have open growth plates (keep their GPC) even after they "stop growing". You even mention this in your discussion.

2) I think you need to add that in your methods that these are all captive bred since I was going to ask the question of captive vs wild until I read the discussion.

So then my question, which you hint at in the discussion, is how does it affect your results to have captive animals? Indeterminate growth and in fact, growth speed/overall metabolic rate in reptiles are highly affected by environmental conditions. If these are zoo animals were raised in less than ideal, which can be very easy to do for exotics, especially those who are very sensitive to living in captivity like some geckos, than this would affect your results. Vica versa, a steady amount of food, low stress and constant temperatures in an ideal zoo setting would speed things up.

Ideally I think you need SOME wild species to compare since reptiles are so plastic in their growth depending on conditions.

Review form: Reviewer 2

Recommendation

Reject – article is scientifically unsound

Scientific importance: Is the manuscript an original and important contribution to its field?

Good

General interest: Is the paper of sufficient general interest?

Good

Quality of the paper: Is the overall quality of the paper suitable?

Poor

Is the length of the paper justified?

Yes

Should the paper be seen by a specialist statistical reviewer?

No

Do you have any concerns about statistical analyses in this paper? If so, please specify them explicitly in your report.

No

It is a condition of publication that authors make their supporting data, code and materials available - either as supplementary material or hosted in an external repository. Please rate, if applicable, the supporting data on the following criteria.

Is it accessible?

Yes

Is it clear?

Yes

Is it adequate?

Yes

Do you have any ethical concerns with this paper?

No

Comments to the Author

The authors examined multiple squamate taxa for the presence of GPC in femora to identify evidence of determinate growth, with implications that determinate growth is an apomorphy of amniote tetrapods. This is an exciting and novel approach for addressing the long-held assumption that lizards and other reptiles can “grow forever”. It is also exciting in that the method provides yet another independent way to identify determinate growth and skeletal maturity in addition to mark-recapture studies and identification of an EFS, especially for the fossil record when a diaphysis might not be available for histologic investigation. In addition, the manuscript provides important evidence for skeletal maturity and asymptotic growth in a range of squamate taxa, and to my knowledge such an extensive investigation of skeletal signals for determinate growth has not previously been performed on this group. I commend the authors for such important research. However, I feel the manuscript requires substantial revisions before it can be published:

- 1) While the results are significant, it seems that the authors themselves missed the true

importance and core finding of their work: that they were in fact testing the hypothesis/assumption of indeterminate growth in tetrapods, and that their results reject this hypothesis. Science is always looking to test and reject hypotheses (in this case indeterminate growth), not evidence to support a hypothesis. But this manuscript is written in the latter form, with the authors trying to find support for determinate growth, and then being surprised that their data shows determinate growth is pervasive. The authors then try to justify/explain how or why they find so many cases of determinate growth when indeterminate growth is “the norm”.

2) The manuscript needs to be rewritten to acknowledge early on that other methods have found evidence for determinate growth in squamates using mark-recapture studies and histology with the EFS, for instance. These other methods are finally discussed briefly in the discussion section, and the introduction is written in a way that makes the authors’ research seem like the first to find good evidence of determinate growth in squamates. In addition, the authors admit to not knowing for certain that skeletal maturity was attained if a GPC is still visible, or at what point growth might cease when the GPC is being obliterated. This makes the use of GPC to confirm or refute indeterminate growth subjective. To not only solidify their argument but also to establish timing, the authors need to include in their revision correlation between the condition of the GPC and the presence or absence of an EFS.

3) Even without the above concerns, the literature cited is not up to date. I have included citations where I think they are needed, but urge the authors to do a thorough literature search of their own as I have only limited time to give to this review.

I’ve made comments directly on the pdf for consideration.

If possible, I would encourage the authors to make the revisions suggested above and resubmit a new manuscript for review.

Decision letter (RSPB-2020-0670.R0)

09-Apr-2020

Dear Professor Frynta:

I am writing to inform you that your manuscript RSPB-2020-0670 entitled "Determinate growth is predominant and likely ancestral in squamate reptiles" has, in its current form, been rejected for publication in Proceedings B.

This action has been taken on the advice of referees, who have recommended that substantial revisions are necessary. With this in mind we would be happy to consider a resubmission, provided the comments of the referees are fully addressed. However please note that this is not a provisional acceptance. Also please note that this decision may (or may not) have taken into account confidential comments.

The reviewers and associate editor agree that the paper deserves further consideration but that it has some core issues needing amending. A reconsideration of hypothesis testing, more literature citations of key studies (and fair acknowledgement of other studies w/similar findings), and more images of CT slices/data in the Supp Info are requested.

The data accessibility statement does not make it clear how much data will be supplied. In your revision process, please take a second look at how open your science is; our policy is that *ALL* (maximally inclusive) data involved with the study should be made openly accessible, fully enabling re-use, replication and transparency-- see: <https://royalsociety.org/journals/ethics-policies/data-sharing-mining/>

Insufficient sharing of data can delay or even cause rejection of a paper. Here, our stringent policy indicates that all original CT data must be supplied, to enable

replication/reuse/repurposing of these valuable data. While such datasets may be huge, repositories such as Dryad or Figshare can handle them with some advance planning and uploading. Ideally, have that ready with your resubmission, giving a private link to reviewers+editor.

Sincerely,

Dr John R Hutchinson, Editor
 mailto: proceedingsb@royalsociety.org

Associate Editor
 Board Member: 1
 Comments to Author:

Thank you for submitting your manuscript to Proceedings B. Both the reviewers find merit in your study, and both commend you on the novel approach taken. However, the reviewers have identified significant issues with the framing of the study and have requested the inclusion of additional data. Please address these issues before considering a resubmission of the manuscript.

Reviewer(s)' Comments to Author:

Referee: 1

Comments to the Author(s)

The authors present a detailed analyses of the Growth Plate Cartilage (GPC) resorption using uCT on a wide range of taxa within the Lepidosauria as a means to see if there is in fact indeterminate growth within this group.

Overall I found their uCT interpretation sound, though I would have liked to see some more images that included some more detailed histology (higher resolution). However, I have two main concerns. The first is their definition of "indeterminate growth" for vertebrates, which MUST be addressed for this paper to make any sense. The second is to talk more about their use of captive animals. See below for more details.

1) What is the definition of indeterminate growth, specifically for vertebrates? From what I had understood, indeterminate growth just meant perpetual growth. Thus this should include both growth plate lengthening AND periosteal enlargements. In fact, most of what I've seen for indeterminate growth in vertebrates follow the later form of perpetual growth that is "width-wise" through the periosteum, which could continue AFTER the closure of the growth plates. In paleohistology, we usually look for the formation of an Outer Circumferal Layer (OCL) to mark the end of growth.

In addition, it would be very messy to use the loss of the growth plate cartilage as your definition of arrested growth as rodents have open growth plates (keep their GPC) even after they "stop growing". You even mention this in your discussion.

2) I think you need to add that in your methods that these are all captive bred since I was going to ask the question of captive vs wild until I read the discussion.

So then my question, which you hint at in the discussion, is how does it affect your results to have captive animals? Indeterminate growth and in fact, growth speed/overall metabolic rate in reptiles are highly affected by environmental conditions. If these are zoo animals were raised in less than ideal, which can be very easy to do for exotics, especially those who are very sensitive to living in captivity like some geckos, than this would affect your results. Vica versa, a steady amount of food, low stress and constant temperatures in an ideal zoo setting would speed things up.

Ideally I think you need SOME wild species to compare since reptiles are so plastic in their growth depending on conditions.

Referee: 2

Comments to the Author(s)

The authors examined multiple squamate taxa for the presence of GPC in femora to identify evidence of determinate growth, with implications that determinate growth is an apomorphy of amniote tetrapods. This is an exciting and novel approach for addressing the long-held assumption that lizards and other reptiles can "grow forever". It is also exciting in that the method provides yet another independent way to identify determinate growth and skeletal maturity in addition to mark-recapture studies and identification of an EFS, especially for the fossil record when a diaphysis might not be available for histologic investigation. In addition, the manuscript provides important evidence for skeletal maturity and asymptotic growth in a range of squamate taxa, and to my knowledge such an extensive investigation of skeletal signals for determinate growth has not previously been performed on this group. I commend the authors for such important research. However, I feel the manuscript requires substantial revisions before it can be published:

1) While the results are significant, it seems that the authors themselves missed the true importance and core finding of their work: that they were in fact testing the hypothesis/assumption of indeterminate growth in tetrapods, and that their results reject this hypothesis. Science is always looking to test and reject hypotheses (in this case indeterminate growth), not evidence to support a hypothesis. But this manuscript is written in the latter form, with the authors trying to find support for determinate growth, and then being surprised that their data shows determinate growth is pervasive. The authors then try to justify/explain how or why they find so many cases of determinate growth when indeterminate growth is "the norm".

2) The manuscript needs to be rewritten to acknowledge early on that other methods have found evidence for determinate growth in squamates using mark-recapture studies and histology with the EFS, for instance. These other methods are finally discussed briefly in the discussion section, and the introduction is written in a way that makes the authors' research seem like the first to find good evidence of determinate growth in squamates. In addition, the authors admit to not knowing for certain that skeletal maturity was attained if a GPC is still visible, or at what point growth might cease when the GPC is being obliterated. This makes the use of GPC to confirm or refute indeterminate growth subjective. To not only solidify their argument but also to establish

timing, the authors need to include in their revision correlation between the condition of the GPC and the presence or absence of an EFS.

3) Even without the above concerns, the literature cited is not up to date. I have included citations where I think they are needed, but urge the authors to do a thorough literature search of their own as I have only limited time to give to this review.

I've made comments directly on the pdf for consideration.

If possible, I would encourage the authors to make the revisions suggested above and resubmit a new manuscript for review.

Author's Response to Decision Letter for (RSPB-2020-0670.R0)

See Appendix A.

RSPB-2020-2737.R0

Review form: Reviewer 1

Recommendation

Accept with minor revision (please list in comments)

Scientific importance: Is the manuscript an original and important contribution to its field?

Good

General interest: Is the paper of sufficient general interest?

Good

Quality of the paper: Is the overall quality of the paper suitable?

Good

Is the length of the paper justified?

Yes

Should the paper be seen by a specialist statistical reviewer?

No

Do you have any concerns about statistical analyses in this paper? If so, please specify them explicitly in your report.

No

It is a condition of publication that authors make their supporting data, code and materials available - either as supplementary material or hosted in an external repository. Please rate, if applicable, the supporting data on the following criteria.

Is it accessible?

Yes

Is it clear?

Yes

Is it adequate?

Yes

Do you have any ethical concerns with this paper?

No

Comments to the Author

Thank you for addressing my comments and suggestions in my first review. I have only a few small questions and comments.

Abstract

Line 21: The line "Body growth is typically indeterminate in ectothermic vertebrates." is misleading for this paper as the point of your paper is to show it is NOT typical. Perhaps add "Body growth is typically THOUGHT to be..."?

Introduction

Line 88: Bone girth is determined by the periosteum AND the endosteum

Lines 105-107: Pterosaurs, a good portion of dinosaurs and all birds are all endothermic, so they might not be the best examples if what you are also trying to show in this paper is "ectothermic does not equally indeterminate growth".

Methods:

Lines 151-155: You say "a relatively dense structure, likely associated with the secondary ossification centre, was typically recognizable in the epiphysis". Were you not able to fully dissect any individuals to confirm this?

Discussion General: You admit that species could be both determinate and indeterminate depending on the environmental factors they grew up in. Given that there have been many harsh periods in geologic time that ancestral lizards would have to face, how would you separate those that are showing an actual character of their species vs those facing difficult environmental stresses?

Decision letter (RSPB-2020-2737.R0)

20-Nov-2020

Dear Professor Frynta

I am pleased to inform you that your manuscript RSPB-2020-2737 entitled "Determinate growth is predominant and likely ancestral in squamate reptiles" has been accepted for publication in Proceedings B. Congratulations!!

The referee(s) have recommended publication, but also suggest some minor revisions to your manuscript. Therefore, I invite you to respond to the referee(s)' comments and revise your manuscript. Because the schedule for publication is very tight, it is a condition of publication that you submit the revised version of your manuscript within 7 days. If you do not think you will be able to meet this date please let us know. As noted, please do include the additional histological thin-section work in the main MS as much as possible.

To revise your manuscript, log into <https://mc.manuscriptcentral.com/prsb> and enter your Author Centre, where you will find your manuscript title listed under "Manuscripts with Decisions." Under "Actions," click on "Create a Revision." Your manuscript number has been appended to denote a revision. You will be unable to make your revisions on the originally

submitted version of the manuscript. Instead, revise your manuscript and upload a new version through your Author Centre.

Sincerely,

Dr John Hutchinson, Editor

Associate Editor

Board Member

Comments to Author:

Thank you for resubmitting your manuscript to Proceedings B. We appreciate the time and consideration the authors have invested in improving the manuscript, and the article is now considerably improved. The first referee now has only minor comments that require some rephrasing or additional clarification. Whilst I appreciate the challenge in correlating an EFS to a resorbed GPC, I would encourage the authors to actually include the additional histological thin-section work they have done in response to Referee 2 into the main body of the MS. I think this work is worth adding.

Reviewer(s)' Comments to Author:

Referee: 1

Comments to the Author(s).

Thank you for addressing my comments and suggestions in my first review. I have only a few small questions and comments.

Abstract

Line 21: The line "Body growth is typically indeterminate in ectothermic vertebrates." is misleading for this paper as the point of your paper is to show it is NOT typical. Perhaps add "Body growth is typically THOUGHT to be..."?

Introduction

Line 88: Bone girth is determined by the periosteum AND the endosteum

Lines 105-107: Pterosaurs, a good portion of dinosaurs and all birds are all endothermic, so they might not be the best examples if what you are also trying to show in this paper is "ectothermic does not equally indeterminate growth".

Methods:

Lines 151-155: You say "a relatively dense structure, likely associated with the secondary ossification centre, was typically recognizable in the epiphysis". Were you not able to fully dissect any individuals to confirm this?

Discussion General: You admit that species could be both determinate and indeterminate depending on the environmental factors they grew up in. Given that there have been many harsh periods in geologic time that ancestral lizards would have to face, how would you separate those that are showing an actual character of their species vs those facing difficult environmental stresses?

Author's Response to Decision Letter for (RSPB-2020-2737.R0)

See Appendix B.

Decision letter (RSPB-2020-2737.R1)

27-Nov-2020

Dear Professor Frynta

I am pleased to inform you that your manuscript entitled "Determinate growth is predominant and likely ancestral in squamate reptiles" has been accepted for publication in Proceedings B.

Open Access

Paper charges

Sincerely,
Editor, Proceedings B
mailto: proceedingsb@royalsociety.org

Appendix A

Revised Manuscript RSPB-2020-0670

Response to Receiving Editor

Dear prof. Hutchinson,

Herewith, we resubmit the revised version of our manuscript entitled: “Determinate growth is predominant and likely ancestral in squamate reptiles”. We appreciate the valuable comments of both reviewers, which have helped us to improve the paper. We have carefully dealt with all the issues raised by the reviewers.

Most importantly, we have thoroughly revised the Introduction and many parts of the Discussion in order to clarify general concepts of indeterminate and determinate growth, explicitly formulate criteria used to assess growth patterns, acknowledge other methodological approaches and earlier evidence for determinate growth in reptiles, and interpret our findings critically. We followed carefully all the reviewers' suggestions concerning changes to the revised text. Moreover, we now clearly specify the origin of the examined individuals in Table S2. Out of 194 individuals examined, 130 were captive bred and 64 wild animals. Literature references have been updated throughout the manuscript. To comply with the maximum length allowed by the journal, we had to move some methodological details and discussion concerning potential technical limitations of the study to the Electronic supplementary material.

All data supporting the analyses performed in this manuscript have been deposited in Dryad. A private link is: <https://datadryad.org/stash/share/VW1yZmu-gvnkszWgcW6cpu4ssut-QFOQKQOfTN8LIes>.

Thus, we believe we have complied with all the reviewers' requests.

Please find below our reply to the reviewers' comments (reviewers' comments are in blue, our replies are in black).

We hope that the manuscript is now acceptable for publication and are looking forward to hearing from you.

Sincerely Yours,

On behalf of all authors,

Daniel Frynta

Response to Reviewer 1

Thank you very much for taking the time to review our paper, its positive evaluation and your constructive comments that helped us to improve the paper. Our responses are detailed below.

The authors present a detailed analyses of the Growth Plate Cartilage (GPC) resorption using μ CT on a wide range of taxa within the Lepidosauria as a means to see if there is in fact indeterminate growth within this group. Overall I found their μ CT interpretation sound, though I would have liked to see some more images that included some more detailed histology (higher resolution).

Due to the size limit for electronic supplementary material, we were able to upload only four sample videos providing examples of the detailed visualizations. Specifically, transversal and frontal μ CT cross-sections of proximal part of femurs showing two specimens of the Moroccan eyed lizard *Timon tangitanus* – a subadult with GPC and an adult without GPC – were uploaded to the Electronic supplementary material (Video File S1-4). While these videos allow slice-by-slice tracking of the GPC in the subadult specimen, they clearly show trabecular bone architecture of the entire metaphysis in the adult specimen.

More importantly, we have deposited all the sequentially acquired μ CT and μ RTG scans of all specimens examined in this and our previous studies in Dryad. A total of 74 GB of data provides information on 361 examined individuals representing 147 species of squamate reptiles. Thus, virtually all data supporting the analyses performed in this manuscript have been deposited in Dryad. At the moment they can be accessed via a private link <https://datadryad.org/stash/share/VW1yZmu-gvnkszWgcW6cpu4ssut-QFOQKQOfTN8LIes>, please feel free to use it for review purposes. After the publication of the paper, this data will be associated with the paper and freely available.

However, I have two main concerns. The first is their definition of “indeterminate growth” for vertebrates, which MUST be addressed for this paper to make any sense. The second is to talk more about their use of captive animals. See below for more details.

1) What is the definition of indeterminate growth, specifically for vertebrates? From what I had understood, indeterminate growth just meant perpetual growth. Thus this should include both growth plate lengthening AND periosteal enlargements. In fact, most of what I’ve seen for indeterminate growth in vertebrates follow the later form of perpetual growth that is “width-wise” through the periosteum, which could continue AFTER the closure of the growth plates. In paleohistology, we usually look for the formation of an Outer Circumferal Layer (OCL) to mark the end of growth. In addition, it would be very messy to use the loss of the growth plate cartilage as your definition of arrested growth as rodents have open growth plates (keep their GPC) even after they “stop growing”. You even mention this in your discussion.

General concepts of indeterminate and determinate growth are addressed in the revised Introduction and in the second paragraph of the Discussion (section Determinate versus

indeterminate body growth). Surprisingly enough, there is no general consensus concerning the definition of indeterminate growth in vertebrates. Because of practical reasons, we have adopted a definition based on survival, postulating that “growth is determinate if an organism reaches maximum (asymptotic) size when many individuals of the population are still alive, and indeterminate, when very few individuals are alive” (Karkach 2006). Such a quantitative distinction is in line with the high frequency of evolutionary transitions between these growth types observed in this study (see section Determinate versus indeterminate body growth for details).

It is important to note that the present study is designed to conclusively identify only cases of determinate growth. In the last paragraph of the Introduction we clearly define what criteria must be met to mark a species as determinate grower: “We consider determinate growers only those species, in which adult individuals exhibit a clear sign of irreversible arrest of longitudinal skeletal growth, i.e., fully resorbed GPCs. Thus, when animals are still capable of growth, although the actual growth rates might be negligible, we consider them potentially indeterminate growers. While this approach may overestimate the number of species exhibiting indeterminate growth, identification of determinate growers is conclusive.” We believe that this restrictive approach provides a highly reliable and transparent way to interpret our data.

As we summarize briefly in the Introduction, different proxies can be used to assess body growth. Since body mass is largely dependent on body condition and significant mass gains can occur even in adults with arrested skeletal growth, long-term growth studies often utilize body length as proxy for body size. This makes a lot of sense because the lengthening of the body through endochondral ossification is the major type of body growth in amniotes (Hall 2005). Osteological studies utilize two different proxies of the arrest of skeletal growth: resorption of the growth plate cartilages (GPCs) as a sign of irreversible arrest of endochondral/longitudinal growth, and the development of the external fundamental system (EFS), also known as Outer Circumferal Layer (OCL), as a sign of cessation of any significant periosteal/circumferential growth of a bone. Due to mechanical reasons and shared molecular regulation, endochondral and periosteal bone growth are coupled and synchronized (Ornitz and Marie 2002, Hall 2005). Thus, we consider both degradation of GPC and the development of EFS to be reliable and largely complementary markers of arrest of skeletal growth. However, both have limitations that need to be considered when interpreting data.

As we clearly indicate in the Discussion (the last paragraph of the section Technical considerations), complete GPC degradation might not precede, but rather follow the cessation of body growth. Rodents may serve as an extreme example of this phenomenon (e.g., Roach et al. 2003, del Pozo et al. 2014), although it has been demonstrated that multiple small bony bridges develop in the rat proximal tibial GPC, which render further longitudinal growth unlikely despite the fact that extensive GPC remains (Martin et al. 2003). In any case, the preservation of GPC is a good proxy for growth capacity, but it does not necessarily indicate actual growth, as the GPC can be present but not active. Hence, as we clearly indicate in the last paragraph of the Introduction, our analysis may overestimate the number of species exhibiting indeterminate growth, as we cannot differentiate cases where growth permanently

stops while the GPC is still at least partly preserved. That is why we have adopted a conservative approach and use the term extended (potentially indeterminate) adult growth throughout the text.

Development of EFS, by contrast, can start already in subadults (e.g., Horner et al. 2000) and marks deceleration rather than cessation of periosteal growth. Thus, it is impossible to tell whether an individual with well-developed EFS already stopped skeletal growth or is still growing at a low rate (e.g., Laurin and de Buffrénil 2016), unless the actual age of the individual is known. The very principle of EFS development, i.e., slow apposition of new layers of lamellar osseous tissue on the surface of the bone diaphysis, requires continuation of slow growth for at least a few seasons after growth deceleration, because EFS is a microstructure of closely spaced series of lines of arrested growth (LAGs), which themselves arise seasonally. It is therefore clear that the development of EFS starts well before cessation of skeletal growth. Importantly, the EFS does not develop when rapid growth stops abruptly, when growth is aseasonal or in short-lived animals. There is also some leeway for subjective judgement about the degree of development of EFS – no clear criteria exist as to how numerous and how tightly spaced the LAGs are supposed to be. Last but not least, it is conceivable that suboptimal fossilization will not preserve LAGs and EFS will be artificially missing in determinate growers.

2) I think you need to add that in your methods that these are all captive bred since I was going to ask the question of captive vs wild until I read the discussion. So then my question, which you hint at in the discussion, is how does it affect your results to have captive animals? Indeterminate growth and in fact, growth speed/overall metabolic rate in reptiles are highly affected by environmental conditions. If these are zoo animals were raised in less than ideal, which can be very easy to do for exotics, especially those who are very sensitive to living in captivity like some geckos, than this would affect your results. Vica versa, a steady amount of food, low stress and constant temperatures in an ideal zoo setting would speed things up. Ideally I think you need SOME wild species to compare since reptiles are so plastic in their growth depending on conditions.

Thank you for drawing our attention to this potentially weak point. Obviously we were unclear about the origin of animals in the first version of the manuscript. Emphasizing the importance of captive animals in the Discussion could even give the impression that wild animals have not been involved in this study at all. In the revised Materials and Methods, we clearly indicate that out of 194 individuals examined, 130 were captive bred and 64 wild animals. In other words, captive bred animals constituted two-thirds and wild animals one third of the examined specimens. The origin of individual samples is labelled in Table S2 as captive vs wild. Importantly, our examination of wild and captive bred individuals yielded consistent results.

It has to be stressed in this context that our study would be impossible without extensive use of captive bred animals, as it was the only way to get fully grown adults (i.e., with larger than 80% SVL_{rel}) of many species. Museum collections are not very helpful in this respect, because they are typically full of subadult and young adult animals, which are not suitable for a study of this type.

Response to Reviewer 2

Thank you very much for evaluation of our work and for a detailed review that helped us to improve the paper significantly. Please find below our reply to your comments.

The authors examined multiple squamate taxa for the presence of GPC in femora to identify evidence of determinate growth, with implications that determinate growth is an apomorphy of amniote tetrapods. This is an exciting and novel approach for addressing the long-held assumption that lizards and other reptiles can “grow forever”. It is also exciting in that the method provides yet another independent way to identify determinate growth and skeletal maturity in addition to mark-recapture studies and identification of an EFS, especially for the fossil record when a diaphysis might not be available for histologic investigation. In addition, the manuscript provides important evidence for skeletal maturity and asymptotic growth in a range of squamate taxa, and to my knowledge such an extensive investigation of skeletal signals for determinate growth has not previously been performed on this group. I commend the authors for such important research.

We greatly appreciate that you are so positive about our manuscript!

However, I feel the manuscript requires substantial revisions before it can be published.

1) While the results are significant, it seems that the authors themselves missed the true importance and core finding of their work: that they were in fact testing the hypothesis/assumption of indeterminate growth in tetrapods, and that their results reject this hypothesis. Science is always looking to test and reject hypotheses (in this case indeterminate growth), not evidence to support a hypothesis. But this manuscript is written in the latter form, with the authors trying to find support for determinate growth, and then being surprised that their data shows determinate growth is pervasive. The authors then try to justify/explain how or why they find so many cases of determinate growth when indeterminate growth is “the norm”.

When we initiated this study, we already knew from the literature and our own earlier studies that some squamate reptiles are determinate growers. Therefore, we decided to test the following hypotheses: H_0 : Indeterminate growth is predominant and ancestral for squamate reptiles, determinate growth evolved several times independently within this clade. H_1 : Determinate growth is predominant and ancestral for squamate reptiles, indeterminate growth evolved several times independently within this clade. The reconstruction of character states falsified the H_0 and provided support for the H_1 . On the basis of these data we can safely conclude that determinate growth predominates in squamate reptiles and that the last common ancestor of Squamata was most likely determinate grower. All other inferences, especially scenarios of the evolution of body growth in amniotes, are based on rather fragmental literature data and should be considered preliminary. Our data can by no means rule out the existence of indeterminate growth in squamates (even less so in amniotes) or bring conclusive evidence concerning growth type in the common ancestor of amniotes. Nevertheless, we feel it important to discuss our results in a broader context, as we hope that the paper will stimulate further osteological research providing better phylogenetic coverage, which will ultimately allow clear conclusions to be drawn.

2) The manuscript needs to be rewritten to acknowledge early on that other methods have found evidence for determinate growth in squamates using mark-recapture studies and histology with the EFS, for instance. These other methods are finally discussed briefly in the discussion section, and the introduction is written in a way that makes the authors' research seem like the first to find good evidence of determinate growth in squamates.

Thank you for drawing our attention to this problem. We did not fully realize that the original version of the Introduction was inappropriately biased in favour of our own results. We have thoroughly revised the Introduction to acknowledge other methodological approaches and earlier evidence for determinate growth in reptiles in the current version of the manuscript.

In addition, the authors admit to not knowing for certain that skeletal maturity was attained if a GPC is still visible, or at what point growth might cease when the GPC is being obliterated. This makes the use of GPC to confirm or refute indeterminate growth subjective. To not only solidify their argument but also to establish timing, the authors need to include in their revision correlation between the condition of the GPC and the presence or absence of an EFS.

As we clearly indicate in the Introduction (for details see also our response to Reviewer 1), the present study is designed to conclusively identify only cases of determinate growth. Fully resorbed GPCs are an unambiguous marker of arrested longitudinal skeletal growth. The preservation of GPCs is a good proxy for growth capacity, but it does not necessarily indicate actual growth, as the GPC can be present but not active. Hence, as we clearly indicate in the last paragraph of the Introduction, our analysis may indeed overestimate the number of species exhibiting indeterminate growth, as we cannot differentiate cases where growth permanently stops while the GPC is still at least partly preserved. Therefore, we have adopted a conservative approach and use the term extended (potentially indeterminate) adult growth throughout the revised text.

It has to be stressed that only long-term growth studies can resolve the above described ambiguity, i.e., confirm that an individual with preserved GPC is actually growing. Screening for the presence of the EFS has a limited potential to solve the issue, because it is impossible to tell whether an individual of unknown age possessing well-developed EFS already stopped skeletal growth or is still growing at a low rate (for detailed argumentation, see our response to Reviewer 1). In other words, while the EFS is an efficient tool to establish timing of growth deceleration, it has very limited capacity to reveal the exact timing of growth cessation in cases when the actual age of an individual is unknown.

Nevertheless, we did try to assess the presence and degree of development of the EFS in selected individuals with known age using both μ CT and classical histology. Unfortunately, the lines of arrested growth (LAGs) were not clearly visible in our μ CT scans (see Figure 1). The difference between densities of growth zones and LAGs is too low to enable visualization by conventional μ CT. Phase-contrast synchrotron tomography (e.g. Sanchez et al. 2012) might perhaps yield better results. Since we do not have access to a synchrotron, we have performed a series of histological examinations. Diaphysis cross sections were stained with Ehrlich's Hematoxylin and examined under bright field illumination, phase contrast, Nomarski interference contrast and in polarized light. However, in the majority of the examined samples, LAGs were not clearly visible (see Figure 2 a–c), which might reflect aseasonal growth

in captive bred individuals. Yet, tightly spaced rings of laminar bone depositions in the outer cortex of the analysed bones (see Figure 2 a–c) actually suggest decelerated or ceased growth. In a few samples, we were able to observe LAGs forming EFS (see Figure 2 d). Because we know that the individual, whose femur is shown on Figure 2d, was at least 7 years old, we can safely conclude that this individual has very likely stopped growing in the fourth or fifth year of life (complete resorption of the femoral GPC is coupled with well-developed EFS in this animal). Importantly, such an inference would not be possible if we did not know the age of the animal.

3) Even without the above concerns, the literature cited is not up to date. I have included citations where I think they are needed, but urge the authors to do a thorough literature search of their own as I have only limited time to give to this review.

Once again, thank you for drawing our attention to this weak point and for suggesting relevant literature. We have made an extensive literature survey and updated references throughout the paper. Indeed, we have found many additional sources of relevant information. Unfortunately, we had to cite these sources rather selectively to avoid excessive lengthening of the paper. Moreover, we had to move some methodological details and discussion concerning potential technical limitations of the study to the Electronic supplementary material in order to comply with the maximum length allowed by the journal.

I've made comments directly on the pdf for consideration. If possible, I would encourage the authors to make the revisions suggested above and resubmit a new manuscript for review.

Thank you for all these helpful comments. We respond briefly to them below (see the text following the figures).

Figure 1. Visualisation of femoral diaphyseal cross-sections by μ CT.

(a) At least 30-year old male of Iraqi spiny-tailed lizard *Saara loricata* (ID 232); (b) At least 24-year old male of Mangrove goanna *Varanus indicus* (ID 214); (c) An old male of Moroccan eyed lizard *Timon tangitanus* (ID 101). Note that lines of arrested growth (LAGs) are not visible on μ CT scans. The concentric circles are an artefact of μ CT scanning and subsequent image reconstruction procedure.

Figure 2. Diaphyseal transverse cross-sections of the femur stained with Ehrlich's Hematoxylin. (a) At least 9-year old female of Yellow-throated plated lizard *Gerrhosaurus flavigularis* (ID 99); (b) At least 5-year old male of Balkan green lizard *Lacerta trilineata major* (ID 640); (c) Nearly 12-year old male of Kuhl's flying gecko *Ptychozoon kuhli* (ID 358); (d) At least 7-year old female of Common leopard gecko *Eublepharis macularius* (ID 187). All these individuals were captive bred and featured complete resorption of the GPC. Note that LAGs are clearly visible in (d) but not in (a–c). White bar in (a–c) marks tightly spaced rings of lamellar bone depositions in the outer bone cortex. Arrows in (d) point to LAGs. Abbreviations: EB: endosteal bone, LAGs: lines of arrested growth.

Response to comments that Reviewer 2 made directly to the PDF manuscript

Line 40: While traditionally described, yes, but this "paradigm" has shifted in more recent years. Only one citation in this list (1-8) is from within the last 22 years, and it refers to indeterminate growth in a teleost (which is not a tetrapod). If the authors want to establish that this paradigm is still pervasive, they need to cite more recent examples after stating that it is "commonly mentioned in textbooks and scientific papers" (lines 41-42).

We updated the references focusing on more recent publications addressing growth in reptiles (e.g., Bajer et al. 2015, Beser et al. 2020, Calsbeek and Irschick 2007, Charnov and Warne 2011, Charnov et al 2001, Lailvaux et al. 2004, Hall 2005, Ortega et al. 2017, Werner et al. 2018, Werner and Griebeler 2017, Hariharan et al. 2016, Tsuboi et al. 2018).

Line 42: If you are to use these citations, please justify their use here. Some refer to studies on fish (which are not tetrapods) and xenarthra- the armadillo in any case was recently shown to have determinate growth via EFS: Heck CT, Varricchio DJ, Gaudin TJ, Woodward HN, Horner JR (2019) Ontogenetic changes in the long bone microstructure in the nine-banded armadillo (*Dasypus novemcinctus*). PLoS ONE 14(4): e0215655. <https://doi.org/10.1371/journal.pone.0215655>. Also, I know this is not citable, but I do have a student working on bone histology of the American opossum, and it does possess an EFS.

We omitted the reference concerning armadillo (Ciancio et al. 2012) here and added some more recent relevant references.

Line 43: "Size" is ambiguous. What is meant here and throughout by that term? If referring to mass, then I'd argue all vertebrates have indeterminate growth. If body length, that would be determinate. If reproductive stage is the cutoff point, then humans have indeterminate growth, for instance.

The Introduction was completely rewritten, so this sentence no longer exists. In the revised Materials and Methods, we clearly state that body length was taken as a proxy for body size.

Line 44: Usually? Please cite a reference for this. As I mention above, humans are most certainly not determinate growers by this criterion.

Sentence no longer exists in the revised Introduction.

Line 45: Is this still a valid/accepted definition for indeterminate growth? The citation is from 1982.

We have elaborated this initial part in the revised Introduction. Definitions provided are now based on papers by Sebens (1987) and Karkach (2006).

Line 47: Then how do you differentiate between the two, if there is in fact a difference? And again, what kind of growth curve are you referring to? The parameters of mass growth curve models mean they approach an asymptote but never reach one, so it is circular to say that an animal has indeterminate growth because of the model when the model is made such that an asymptote is not attainable. This is true of any animal modeled using these methods (von Bertalanffy, Gompertz, Logistic).

In the revised Introduction we are more specific about this issue. We argue as follows: "...To complicate matters further, indeterminate growers may exhibit asymptotic growth, provided that the environment imposes energetic constraints on their growth, and determinate growers do not have to achieve asymptotic growth due to high mortality in natural populations. Thus, practical distinction between these two types of growth may be difficult on the basis of empirical growth curves alone, especially in cases when indeterminate growth greatly decelerates with age and/or determinate

growth persists after sexual maturity and its rate is modulated by the environment [2, 3]. Therefore, it is desirable to validate long-term growth studies with osteological examinations, which not only have the potential to reveal irreversible arrest of skeletal growth and thereby provide conclusive evidence for determinate growth, but also can do so relatively quickly across large numbers of species representing various clades.”

Line 53: Are the secondary centers of ossification in endochondral bones of diapsids homologous to those in synapsids?

There are two views on the evolution of secondary ossification centers in Tetrapods reported in the literature. Haines (1942) assumed the evolution of secondary centers from a single origin, while Walter (1985) concluded that: “secondary centers of ossification evolved independently in mammals, snakes and lizards.” A single origin of secondary ossification centers and their secondary loss in turtles and crocodiles is more parsimonious. We decided to omit this unnecessary information concerning the fusion of primary and secondary ossification centers.

Line 55: But there are other "unambiguous signs" as well that should be mentioned here instead of first mentioned in the discussion, such as mark-recapture studies that approach an asymptote and presence of an EFS/OCL. The present study is not the first definitive way to establish determinate growth but another independent test.

We have thoroughly revised the Introduction to acknowledge other methodological approaches and earlier evidence for determinate growth in reptiles in the current version of the manuscript.

Line 59: Again, the present study is an independent way to demonstrate determinant growth. It was not the first to do so, or the first method to do so. What this study does is actually test the hypothesis that indeterminate growth in tetrapods is "real" or simply an assumption for long-lived, slow growing taxa. Prior to this study, the idea of indeterminate growth was not "universal" for squamates, as some of your references demonstrate as well as a few additional examples:

de Buffrénil, V., and J. Castanet. 2000. Age estimation by skeletochronology in the Nile monitor (*Varanus niloticus*), a highly exploited species. *Journal of Herpetology* 34: 414-424.

Botha AE, Botha J. 2019. Ontogenetic and inter-elemental osteohistological variability in the leopard tortoise *Stigmochelys pardalis*. *PeerJ* 7:e8030 <https://doi.org/10.7717/peerj.8030>

Bronikowski, A. M., and S. J. Arnold. 1999. The evolutionary ecology of life history variation in the garter snake *Thamnophis elegans*. *Ecology* 80: 2314-2325.

Lailvaux, S. P., A. Herrel, B. VanHooydonck, J. J. Meyers, and D. J. Irschick. 2004. Performance capacity, fighting tactics and the evolution of life-stage male morphs in the green anole lizard (*Anolis carolinensis*). *Proceedings of the Royal Society of London Series B* 271: 2501-2508.

Congdon, J. D., R. D. Nagle, O. M. Kinney, and R. C. van Loben Sels. 2001. Hypotheses of aging in a long-lived vertebrate, Blanding's turtle (*Emydoidea blandingii*). *Experimental Gerontology* 36: 813-827.

As already mentioned above, we acknowledge other methodological approaches and earlier evidence for determinate growth in reptiles in the revised version of the Introduction. Nevertheless, we are convinced that the universality of indeterminate body growth in reptiles remains very popular, particularly among researchers studying ecology, life-history and behavior of reptiles (e.g., Bajer et al. 2015, Beser et al. 2020, Chamberlain et al. 2017). By the way, two of the suggested papers (Bronikowski and Arnold 1999, Lailvaux et al. 2004) actually also interpret reptile growth as indeterminate, one does not deal with this issue at all (de Buffrénil and Castanet 2000).

Line 66: Please provide examples of where indeterminate growth is well-documented in squamates, and not possibly the result of 1) animals dying before reaching asymptotic body lengths or 2) studies not going on long enough to document asymptotic body length. The conclusion that indeterminate growth evolved independently several times does not make sense in light of your (and others') findings. Instead of assuming the long-held presumption that indeterminate growth is valid in vertebrates, your study helps test that assumption and provides more evidence to reject it. If determinate growth is "predominant", and evidence for indeterminate growth is shaky at best, rather than try to explain the presence of indeterminate growth as having evolved independently, it is more parsimonious to conclude indeterminate growth is not an accurate description for tetrapod growth.

As we indicate in the Introduction, we have adopted a definition of growth types in vertebrates that is based on survival, postulating that "growth is determinate if an organism reaches maximum (asymptotic) size when many individuals of the population are still alive, and indeterminate, when very few individuals are alive" (Karkach 2006). This approach indeed stresses quantitative rather than qualitative differences between growth patterns (see also the section Determinate versus indeterminate body growth for further discussion). According to these criteria, groups in which we consistently observe preserved GPC in adults, namely chameleons, agamas, large species of monitor lizards and teiids, are indeterminate growers. The hypothetical possibility that they all would have preserved but inactive GPCs is highly unlikely.

Line 155: Again, this result should not be surprising based on other methods from the literature supporting determinate growth in squamates.

We omit the word surprisingly.

Lines 163-164: Again, it is most parsimonious to conclude that indeterminate growth is not valid, rather than it having occasionally evolved for some inexplicable reason.

We respectfully do not agree. Published osteological papers (de Buffrénil et al. 2005, Frydlova et al. 2017, 2019) as well as the current study consistently support the existence of extended (potentially indeterminate) growth in the above mentioned reptile groups. To our knowledge, no empirical data that would challenge the existence of indeterminate growth in these reptiles currently exist. Thus, the opinion that indeterminate growth is not valid is not grounded in evidence and remains purely hypothetical.

Lines 170-172: The question may be unanswered because indeterminate growth is not a real phenomenon. The fact that squamate growth is highly plastic also contributes to the assumption that growth can be indeterminate.

As we have argued above, we disagree with the reviewer on this point.

Line 176: If the GPC disappears, and that is your definition of determinate growth, then you have demonstrated that these taxa are determinate growers.

That is a good point. We have reformulated the statement as follows: "The very fact that the GPC disappears in extremely senescent individuals (e.g., 24-year-old *Varanus indicus* [34]; 30-year-old *Uromastyx loricatus* [35]) belonging to reptile groups that otherwise feature extended (potentially indeterminate) adult growth is also in line with the hypothesis that it is the timing that sets apart determinate from indeterminate growth, at least in squamate reptiles."

Line 182: Why not, please explain? And if this is the case, isn't it possible that indeterminate growth is a result of studies not going on long enough, rather than being an actual growth "strategy"?

To provide the requested explanation, we have elaborated the paragraph as follows: “Long-term studies are the only way to accumulate quantitative data on individual growth. However, it is challenging to maintain these studies over sufficiently long time periods to encompass the natural lifespan of individuals, especially in large reptiles that tend to be long-lived. Consequently, long-term growth data is available only for a limited number of species (e.g., [29, 41-44, 46, 77, 78]) and do not allow the reconstruction of growth type evolution across reptiles. In the absence of these data, osteological evidence is invaluable.”

Line 191: In these instances, please explain why the presence of GPC in agamas implies indeterminate growth, if growth curves show an asymptote is approached? Study of the tuatara, which you agree is a determinate grower, shows it takes 15-25 years to totally obliterate the growth plate, so might this also be the case for agama?

The important distinction here is that we do not use asymptotic growth curves as indicators of determinate growth. Growth generally slows down considerably in older individuals, but so long as they are still able to grow, they are not considered determinate growers under our definition. Moreover, if it generally took that long for the GPC to disappear, we would not be able to show its absence in the majority of species investigated. We think the timing of GPC resorption in the tuatara is linked to its extreme longevity and not a general phenomenon.

Lines 194-195: Again, "as previously thought"- does this go back to the citations in the introduction? Because all but one were over 20 years old and more evidence of determinant growth is presented more recently.

We have reformulated the sentence as follows: “Together, these results suggest that determinate growth is widespread among extant lizard species.”

Lines 197-199: Please also cite the histological evidence for determinate growth: Castanet, J., D. G. Newman, and H. S. Girons. 1988. Skeletochronological data on the growth, age, and population structure of the tuatara, *Sphenodon punctatus*, on Stephens and Lady Alice Islands, New Zealand. *Herpetologica* 44: 25-37.

This has been cited in the submitted manuscript (reference no. 10) as it is in the revised one (reference no. 28).

Lines 199-200: But according to Castanet et al. 1988, it takes 15-25 years for tuatara to obliterate the GPC, so if you were to look for GPC in a tuatara and found one, you might conclude it to have indeterminate growth? Castanet, J., D. G. Newman, and H. S. Girons. 1988. Skeletochronological data on the growth, age, and population structure of the tuatara, *Sphenodon punctatus*, on Stephens and Lady Alice Islands, New Zealand. *Herpetologica* 44: 25-37.

Yes, this is completely true. However, it is important to link the timing of GPC resorption with longevity. According to Castanet et al. 1988, GPC is resorbed at the age of 15-25 years. However, for tuatara this is the first third of its life due to the amazing longevity of this species. The animal still has most of its reproduction span and lifetime ahead. Thus, we concluded that this is a case of determinate body growth. On the other hand, GPC resorption in *Varanus indicus* or *Uromastix loricatus* is completely different. GPC resorption appeared at a highly senescent age.

Line 209: To confirm the significance of the persistence of the GPC and whether that implies indeterminate growth, it would have been good for you to supplement your CT data with histological sections transversely through midshaft to establish presence or absence of a corresponding EFS. You could likely also see this via micro CT. A recent study on red deer examined the timing of epiphyseal

fusion and appearance of EFS to do just this, showing the two coincided. Until the GPC is correlated with an EFS (or not), it is impossible to say at what point growth stops when a GPC is still visible.

As we have argued in detail above, conventional μ CT is not suitable for assessment of EFS presence/absence, likely due to low difference between densities of growth zones and LAGs. Likewise, our histological examinations did not bring satisfactory results. In the majority of the examined samples, LAGs were not clearly visible, which might reflect aseasonal growth in captive bred individuals. Moreover, the material used in this study is not suitable for analysis of the timing of skeletal maturation. Complete ontogenetic series of a model species should be used for this purpose. We actually plan to perform such an analysis in a follow-up study.

Besides, histological examination of all species involved in the character state reconstruction analysis is clearly beyond the scope of this paper. In fact, there are only a few published papers combining these two techniques and none of them analyse more than a handful of species. A vast majority of papers apply only one technique in one or a few species. Limitations of both techniques are addressed in detail above (see response to Reviewer 1).

Lines 211 a 215: EFS has also been found in Nile monitors, and in turtles (now considered archisauriformes):

de Buffr nil, V., and J. Castanet. 2000. Age estimation by skeletochronology in the Nile monitor (*Varanus niloticus*), a highly exploited species. *Journal of Herpetology* 34: 414-424.

Botha AE, Botha J. 2019. Ontogenetic and inter-elemental osteohistological variability in the leopard tortoise *Stigmochelys pardalis*. *PeerJ* 7:e8030 <https://doi.org/10.7717/peerj.8030>

We now cite the paper by Botha and Botha (2019), thanks for the suggestion. However, we did not find a clear evidence for EFS in the paper by de Buffr nil and Castanet (2000). Therefore, we refer to a later paper by the same author (de Buffr nil et al. 2005).

Lines 224-226: This citation is from 2010. Much has been done on early synapsids in the past 10 years, and there is now evidence of determinate growth. Ophiacodon was reported to have an EFS in 2017: Christen D. Shelton, Paul Martin Sander, Long bone histology of Ophiacodon reveals the geologically earliest occurrence of fibrolamellar bone in the mammalian stem lineage, *Comptes Rendus Palevol*, Volume 16, Issue 4, 2017. And *Varanops* was just reported to have an EFS (OCL) in Adam K. Huttenlocker and Christen D. Shelton. 2020. Bone histology of varanopids (Synapsida) from Richards Spur, Oklahoma, sheds light on growth patterns and lifestyle in early terrestrial colonizers *Phil. Trans. R. Soc.*

Thank you for bringing this issue to our attention and for the suggested literature. We have carefully screened the current literature and updated the paragraph concerning synapsids accordingly. Please note that *Varanops* was reported not to have an EFS by Huttenlocker and Shelton (2020).

Line 233: Citation needed.

We provided a reference as requested.

Lines 249-251: This again assumes indeterminate growth is "real"- consider that you are finding evidence that rejects the assumption.

Once again, we do believe that indeterminate growth is a real phenomenon and this opinion is supported by empirical evidence. We are not aware of any evidence supporting the notion that indeterminate growth is a methodological artefact.

Line 272: Please explain why it would not be affected? Reptiles are renowned for their plasticity in growth due to external factors. We have no idea how long the GPC would stick around in an animal raised in optimal conditions versus one taking twice as long to get to adult size in the wild, for instance. That's why correlating GPC with EFS is so critical for accurate interpretation.

It is true that the growth rate may differ due to external factors, especially in squamates with high plasticity of body growth. Nevertheless, external factors are able to postpone/enhance body growth (consider e.g. the difference among spring and summer clutch), but the general mode of body growth will be preserved, because it is genetically determined. This was proved in laboratory experiments in *Paroedura picta* (Kubička and Kratochvíl 2009). Low-fed and well-fed females followed the same growth trajectory and attained similar body lengths (SVL) regardless of the amount of available food. On the other hand, allocation to reproduction and body condition (the amount of body fat) was highly nutrition-dependent.

Concerning the EFS, we think that this will bring no additional information, as it has been demonstrated that GPC degradation tends to be synchronized with EFS development.

Lines 277-278: Earlier you said the GPC cannot be taken as evidence of indeterminate growth, but here you claim it overestimates indeterminate growth.

While it is true that the preservation of GPC cannot be taken as a conclusive evidence, it certainly provides support for indeterminate growth. Strictly speaking, it is evidence of the potential to grow.

We have reformulated the paragraph to clarify this issue. Briefly, we apply a restrictive definition of determinate body growth. Only entirely absent GPC is taken as a reliable indicator of arrested growth and thereby an indicator of determinate growth. All other character states, including partly resorbed GPC, are scored as GPC present and taken as an indicator of potential indeterminate growth.

Lines 278-280: A publication in 2019 correlated epiphyseal fusion with the appearance of an EFS in red deer. I think this paper and its implications to the present study should be discussed: Calderón, Teresa, et al. "Calibration of life history traits with epiphyseal closure, dental eruption and bone histology in captive and wild red deer." *Journal of anatomy* 235.2 (2019): 205-216.

Thank you for this suggestion. It is a very useful reference.

Lines 285-286: What your study really does is reject the hypothesis of indeterminate growth.

We do not agree with this interpretation. Instead, our study brings evidence, albeit at times inconclusive, for independent evolutionary origin of extended (potentially indeterminate) adult body growth in several phylogenetically distant groups of reptiles.

References

- Bajer, K., G. Horvath, O. Molnar, J. Torok, L. Z. Garamszegi, and G. Herczeg (2015). European green lizard (*Lacerta viridis*) personalities: Linking behavioural types to ecologically relevant traits at different ontogenetic stages. *Behavioural Processes* 111:67-74.
- Beser, N., C. Ilgaz, Y. Kumlutas, K. Candan, O. Guclu, and N. Uzum (2020). Age and growth in two populations of Danford's lizard, *Anatololacerta danfordi* (Gunther, 1876), from the eastern Mediterranean. *Turkish Journal of Zoology* 44:173-180.
- Castanet, J., D. G. Newman, and H. Saintgiron (1988). Skeletochronological data on the growth, age, and population-structure of the Tuatara, *Sphenodon punctatus*, on Stephens island and Lady-Alice island, New Zealand. *Herpetologica* 44:25-37.

- De Buffrenil, V., I. Ineich, and W. Bohme (2005). Comparative data on epiphyseal development in the family Varanidae. *Journal of Herpetology* 39:328-335.
- del Pozo, E., M. Janner, A. R. Mackenzie, S. Arampatzis, A. K. Dixon, R. Perrelet, W. Ruch, K. Lippuner, J. Zapf, S. W. Lamberts, and P. E. Mullis (2014). Radiometrical, hormonal and biological correlates of skeletal growth in the female rat from birth to senescence. *Growth Hormone & Igf Research* 24:83-88.
- Frydlova, P., J. Mrzilkova, M. Seremeta, J. Kremen, J. Dudak, J. Zemlicka, P. Nemec, P. Velensky, J. Moravec, D. Koleska, V. Zahradnickova, et al. (2019). Universality of indeterminate growth in lizards rejected: the micro-CT reveals contrasting timing of growth cartilage persistence in iguanas, agamas, and chameleons. *Scientific Reports* 9:14.
- Frydlova, P., V. Nutilova, J. Dudak, J. Zemlicka, P. Nemec, P. Velensky, T. Jirasek, and D. Frynta (2017). Patterns of growth in monitor lizards (Varanidae) as revealed by computed tomography of femoral growth plates. *Zoomorphology* 136:95-106.
- Haines, R. W. (1942). The evolution of epiphysis and of endochondral bone. *Biological Review* 17:267-292.
- Hall, B. K. (2005). *Bones and cartilage: developmental and evolutionary skeletal biology*. Elsevier Academic Press, Australia ; San Diego, Calif.
- Horner, J. R., A. De Ricqles, and K. Padian (2000). Long bone histology of the hadrosaurid dinosaur *Maiasaura peeblesorum*: Growth dynamics and physiology based on an ontogenetic series of skeletal elements. *Journal of Vertebrate Paleontology* 20:115-129.
- Chamberlain, J. D., I. T. Clifton, and M. E. Gifford (2017). Influence of prey size on reproduction among populations of Diamond-backed Watersnakes (*Nerodia rhombifer*). *Canadian Journal of Zoology* 95:929-935.
- Karkach, A. S. (2006). Trajectories and models of individual growth. *Demographic Research* 15:348-+.
- Kubicka, L., and L. Kratochvil (2009). First grow, then breed and finally get fat: hierarchical allocation to life-history traits in a lizard with invariant clutch size. *Functional Ecology* 23:595-601.
- Laurin, M., and V. de Buffrenil (2016). Microstructural features of the femur in early ophiacodontids: A reappraisal of ancestral habitat use and lifestyle of amniotes. *Comptes Rendus Palevol* 15:115-127.
- Martin, E. A., E. L. Ritman, and R. T. Turner (2003). Time course of epiphyseal growth plate fusion in rat tibiae. *Bone* 32:261-267.
- Ornitz, D. M., and P. J. Marie (2002). FGF signaling pathways in endochondral and intramembranous bone development and human genetic disease. *Genes & Development* 16:1446-1465.
- Roach, H. I., G. Mehta, R. O. C. Oreffo, N. M. P. Clarke, and C. Cooper (2003). Temporal analysis of rat growth plates: Cessation of growth with age despite presence of a physis. *Journal of Histochemistry & Cytochemistry* 51:373-383.
- Walter, L. R. (1985). The formation of secondary centers of ossification in Kannemeyrid dicynodonts. *Journal of Paleontology* 59:1486-1488.

Appendix B

Revised Manuscript RSPB-2020-2737

Response to Receiving Editor

Dear prof. Hutchinson,

Herewith, we resubmit the accepted manuscript with a few additional changes requested by the associated editor and the referee 1. Specifically, we have included results of histological examinations of bone diaphysis to Electronic supplementary material and reworded the manuscript in several places to make it clearer. In addition, we have simplified the structure of data deposited in Dryad (the sequential μ CT and μ RTG scans were replaced by video files) to make it more easily accessible to readers.

Thus, we believe we have complied with all requests.

Please find below our reply to the reviewers' comments (reviewers' comments are in blue; our replies are in black) and final main document with tracked changes.

Sincerely Yours,

On behalf of all authors,

Daniel Frynta

Response to Associate Editor

Whilst I appreciate the challenge in correlating an EFS to a resorbed GPC, I would encourage the authors to actually include the additional histological thin section work they have done in response to Referee 2 into the main body of the MS. I think this work is worth adding.

In order to comply with the maximum length of the manuscript allowed by the journal and because of the preliminary nature of the data, we have included the results of pilot histological examinations in the Electronic supplementary material and refer to this data briefly in the Results.

Response to Reviewer 1

Abstract: Line 21: The line “Body growth is typically indeterminate in ectothermic vertebrates.” is misleading for this paper as the point of your paper is to show it is NOT typical. Perhaps add “Body growth is typically THOUGHT to be...”?

Thank you for raising this point. We reworded the first sentence of the abstract as suggested.

Introduction: Line 88: Bone girth is determined by the periosteum AND the endosteum

We have reformulated this part to acknowledge the role of the endosteum in bone growth.

Lines 105-107: Pterosaurs, a good portion of dinosaurs and all birds are all endothermic, so they might not be the best examples if what you are also trying to show in this paper is “ectothermic does not equally indeterminate growth”.

We do not speak specifically about ectotherms in this part of the manuscript but list all species/groups of reptiles for which there is evidence for determinate growth, be they ectotherms or endotherms. Decoupling between endothermy and determinate body growth is addressed in detail in the Discussion.

Methods: Lines 151-155: You say “a relatively dense structure, likely associated with the secondary ossification centre, was typically recognizable in the epiphysis”. Were you not able to fully dissect any individuals to confirm this?

In fact, we are almost sure that the dense structure inside the epiphysis is a secondary ossification centre. However, because we did not verify this histologically, we prefer the conservative statement “likely associated with secondary ossification centre”.

We rephrased the sentence as follows to make it clear:” The presence of the GPC was clearly visible on μ RTG and μ CT scans as fine-grained cartilage and a suture between the metaphysis and epiphysis; a relatively dense structure, likely associated with the secondary ossification centre, was typically recognizable **inside** the epiphysis.”

Discussion General: You admit that species could be both determinate and indeterminate depending on the environmental factors they grew up in. Given that there have been many harsh periods in geologic time that ancestral lizards would have to face, how would you separate those that are showing an actual character of their species vs those facing difficult environmental stresses?

This misunderstanding. We show that evolutionary transitions between determinate and indeterminate growth (and vice versa) are frequent in squamate reptiles. We also point out that environmental conditions may influence actual growth rates, however, we do not challenge the existence of species-specific growth type and certainly do not suggest that growth type is dependent on the environmental conditions animals grow up in. On the contrary, our data show that growth type is largely phylogenetically determined. Life history traits (namely, higher fitness of larger individuals) rather than environmental factors likely account for repeated occurrence of indeterminate growth in large species of monitor lizards and teiids.